# Joint Particle Swarm Optimization of Power and Phase Shift for IRS-Aided D2D Underlaying Cellular Systems

**DOI:** 10.3390/s23115266

**Published:** 2023-06-01

**Authors:** Ruijie Wang, Xun Wen, Fangmin Xu, Zhijian Ye, Haiyan Cao, Zhirui Hu, Xiaoping Yuan

**Affiliations:** 1Key Laboratory of Data Storage and Transmission Technology of Zhejiang Province, Institute of Communications Engineering, Hangzhou Dianzi University, Hangzhou 310018, China; 19081526@hdu.edu.cn (R.W.); 19201130@hdu.edu.cn (X.W.); 202080124@hdu.edu.cn (Z.Y.); caohy@hdu.edu.cn (H.C.); huzhirui@hdu.edu.cn (Z.H.); 2Information Engineering School, Hangzhou Dianzi University, Hangzhou 310018, China; xpyuan@hdu.edu.cn

**Keywords:** D2D communication, IRS, rate optimization, particle swarm optimization (PSO)

## Abstract

Device-to-device (D2D) communication is a promising wireless communication technology which can effectively reduce the traffic load of the base station and improve the spectral efficiency. The application of intelligent reflective surfaces (IRS) in D2D communication systems can further improve the throughput, but the problem of interference suppression becomes more complex and challenging due to the introduction of new links. Therefore, how to perform effective and low-complexity optimal radio resource allocation is still a problem to be solved in IRS-assisted D2D communication systems. To this end, a low-complexity power and phase shift joint optimization algorithm based on particle swarm optimization is proposed in this paper. First, a multivariable joint optimization problem for the uplink cellular network with IRS-assisted D2D communication is established, where multiple DUEs are allowed to share a CUE’s sub-channel. However, the proposed problem considering the joint optimization of power and phase shift, with the objective of maximizing the system sum rate and the constraints of the minimum user signal-to-interference-plus-noise ratio (SINR), is a non-convex non-linear model and is hard to solve. Different from the existing work, instead of decomposing this optimization problem into two sub-problems and optimizing the two variables separately, we jointly optimize them based on Particle Swarm Optimization (PSO). Then, a fitness function with a penalty term is established, and a penalty value priority update scheme is designed for discrete phase shift optimization variables and continuous power optimization variables. Finally, the performance analysis and simulation results show that the proposed algorithm is close to the iterative algorithm in terms of sum rate, but lower in power consumption. In particular, when the number of D2D users is four, the power consumption is reduced by 20%. In addition, compared with PSO and distributed PSO, the sum rate of the proposed algorithm increases by about 10.2% and 38.3%, respectively, when the number of D2D users is four.

## 1. Introduction

High frequency band technologies such as millimeter waves are considered as effective technologies to solve the shortage of spectrum resources. However, in some harsh wireless environments, the scattering of signal propagation is very poor, which seriously affects the millimeter wave communication. Therefore, a low power sustainable network which can control the signal propagation environment becomes an important direction of future communication development. In recent years, intelligent reflective surfaces (IRS) have been listed as a potential key technology for the sixth generation (6G) due to its advantages in regulating the wireless environment, improving signal transmission quality, and its high energy efficiency (EE) [1,2,3,4].

Research shows that IRS has the ability to reduce the impact of environmental obstacles on communication quality through beamforming and phase shift optimization. For example, in [5], the combined optimization of the active beamforming of the base station and the intelligent beamforming of IRS is used to maximize the weighted sum rate of the multiple user Multiple Input Single Output (MISO) downlink system assisted by IRS. In [6], aiming at the IRS-assisted multiple input multiple output (MIMO) system, a deep learning method is adopted to jointly optimize the transmitting beam of the base station and the phase shift of the reflecting element on IRS, so as to maximize the total rate of the IRS-assisted MIMO system. Authors in [7] conduct continuous digital beamforming and discrete IRS-based analog beamforming on BS and IRS, verifying that an IRS-based system can achieve good sum rate performance by setting a reasonable IRS size and a small amount of discrete phase shift. In the literature, the author of [8] integrates IRS into Unmanned Aerial Vehicle (UAV) communication to improve the sum rate of single-antenna UAV systems.

Most of these studies have focused on IRS-assisted multi-user multi-input systems, using phase shift optimization or beamforming to suppress interference and achieve better sum rate performance, but they have not considered IRS-assisted Device-to-device (D2D) communication systems. When there are sufficient spectrum resources in the network and D2D communication adopts the cellular mode, it is not necessary to introduce reflective devices to increase the effective signal. However, when D2D communication users choose the multiplexing mode, especially in the case of high-frequency reuse, it is crucial to use IRS to regulate the wireless propagation environment. Therefore, in reference [9], the authors introduce IRS into a D2D communication system and optimize the power allocation and phase shift adjustment according to block coordinate descent and the semi-relaxation algorithm. As expected, the IRS-assisted D2D communication system achieves a higher sum rate than traditional D2D systems. Reference [10] analyzes the security performance and outage probability of IRS-assisted D2D communication when the direct link is blocked by an obstacle. First, the analysis results of the security outage probability and non-zero security capacity probability of cellular networks are given. Then, the D2D outage probability is also derived. The simulation result shows that the proposed algorithm enhances the secrecy performance effectively. Sun et al., designed an alternate optimization algorithm to minimize the total computational latency by jointly optimizing the computational task allocation, transmitted power control, channel assignment, and phase beamforming. The task allocation strategy is derived by closed-form expression, and the semi-definite relaxation (SDR) method is used to optimize the phase beamforming. As a result, the computation delay is effectively reduced [11]. In [12], authors alternately optimized the transmit power and passive beamforming vector. Firstly, the passive beamforming at the IRS is decoupled based on Lagrange dual transformation, and optimized by fractional programming, and then the Dinkelbach method is used to solve the power optimization sub-problem. According to the relative channel strength, Ref. [13] jointly designs the beamforming, transmit power, and frequency reuse factor of IRS-assisted D2D communication to maximize the EE and spectral efficiency (SE). Specifically, an efficient and low-complexity user pairing scheme is proposed to determine the frequency reuse factor. Then, based on alternating optimization, Lagrange dual transformation, and quadratic transformation techniques, other variables are optimized by the iterative algorithm to maximize SE. Finally, Dinkelbach method is used to solve EE maximization problem. In [14], a two-phase protocol combining IRS and D2D communication was proposed. The advantage of this protocol is that the reliability of communication between the AP and the actuator is doubled through the IRS-assisted first level transport and the second level D2D transport. The joint optimization problem of active and passive beamforming was also proposed with the goal of maximizing the number of successful decoders. Simulation results verify the effectiveness of the algorithm and the effect of IRS on performance improvement. In [15], the authors investigated an IRS-aided millimeter wave D2D communication to enhance the system throughput. Specifically, the millimeter wave D2D communication system model based on IRS was established. Then, the problem of maximizing effective transmission rate was transformed into the IRS activation and selection (IAS) problem, and an approximate solution was proposed to effectively solve the IAS problem. Ref. [16] studied the resource allocation for IRS-assisted joint processing coordinated multipoint downlink cellular networks based on D2D communication, in which IRS is used to mitigate the co-channel interference caused by D2D devices. To maximize the sum rate, a joint optimization was proposed, which focuses on cellular user association, beamforming at the base station, passive beamforming at the IRS, and transmit power allocation. In [17], the authors investigated a joint resource optimization for IRS-D2D communication systems based on game theory. First, the optimization problem of sub-channel allocation, and power and phase shift allocation was established, which proved to be NP-hard. Then, the authors decomposed the optimization problem into three sub-problems and optimized them, respectively, which effectively improved the sum rate of the system. Although the above works significantly improved the total system rate of IRS-assisted D2D communication, they mainly adopted optimization with high complexity. Therefore, it is of great importance to design a resource allocation strategy with high performance and low complexity for the high-reuse scenario of D2D communication assisted by IRS.

Motivated by the advantages of the particle swarm optimization algorithm in solving complex optimization problems, this paper studies the optimization of resource allocation in nonlinear and non-convex IRS-assisted D2D communication systems by swarm intelligence schemes. We propose a low-complexity and high-performance PSO-based solution for the IRS-D2D systems. In order to further evaluate the performance of the proposed algorithm, we analyze the performance of the algorithm in terms of sum rate, power consumption, and convergence by simulation. The contributions of this work can be summarized as follows:(1)To optimize the sum rate for IRS-assisted D2D communication underlying cellular systems, we establish an optimal function that takes into account power allocation and phase shift allocation while satisfying transmission power range and minimum data rate constraints;(2)The proposed multivariable optimization problem is non-convex and nonlinear, so it is difficult to solve directly. To this end, we propose a low-complexity PSO-based wireless radio resource allocation. Different from the existing work, the proposed method does not need to decompose the optimization problem into multiple subproblems and optimize them separately, which avoids high complexity. In addition, unlike previous applications of PSO, in this paper, the PSO algorithm is used to optimize the transmit power and phase shift simultaneously, which has not been covered before;(3)We analyze the performance of the proposed algorithm in terms of sum rate, power consumption, and convergence through simulations to verify its effectiveness. In addition, we provide a complexity analysis.

The remainder of this paper is organized as follows. In Section 2, the system model is described, along with the problem formulation. Section 3 focuses on the joint optimization scheme based on PSO, which is used to solve the optimization problem. Section 4 gives the detailed simulation results. Section 5 concludes the paper. Finally, Section 6 discusses the future work.

## 2. System Model and Problem Description

IRS can effectively improve the performance of the communication system by changing the signal propagation environment. Therefore, IRS-assisted communication systems have received much attention. However, how to effectively allocate radio resources to further improve system performance and reduce co-channel interference remains to be solved.

In order to investigate the wireless resource allocation strategy of IRS-D2D communication systems, we first establish the system model in this section. Then, by analyzing the signal to interference plus noise ratio (SINR), the objective function and constraint conditions of optimization are proposed. In general, wireless resource management for IRS-assisted systems includes channel allocation, power control, and phase shift allocation. Here, we assume that channel allocation is complete, and focus on power and phase shift allocation. That is to say, we consider the power and phase shift optimization among users sharing the same channel after the completion of sub-channel allocation. As shown in Figure 1, the IRS-assisted D2D communication system has a BS, an IRS containing N modules placed near the base station, and I users, including a cellular user and I−1 pairs of co-channel D2D users.

In the system, there is co-channel interference between any two users: the cellular user is interfered with by D2D users, and D2D users are also interfered with by the cellular user and other D2D users. Therefore, any user i in the system will receive interference from a user $j\left(j\neq i\right)$.

Here, $h_{r_{i},t_{i}}$ is the channel gain of the direct link between the transmitter $t_{i}$ and the desired receiver $r_{i}$, and $g_{r_{i},t_{i}}^{n}$ is the channel gain of the reflection link reflected from transmitter $t_{i}$ to receiver $r_{i}$ by the *n*th reflection module. Then, the signal to interference plus noise ratio (SINR) of user i can be expressed as
(1)$$\mathrm{SINR}=\frac{{\left|h_{r_{i},t_{i}}+\sum_{n=1}^{N}g_{r_{i},t_{i}}^{n}e^{j\theta_{n}}\right|}^{2}p_{i}}{\sum_{j=1,j\neq i}^{I}{\left|h_{r_{i},t_{j}}+\sum_{n=1}^{N}g_{r_{i},t_{j}}^{n}e^{j\theta_{n}}\right|}^{2}p_{j}+𝜛}$$
where $p_{i}$ and $p_{j}$ represent the transmission power from the transmitter $t_{i}$ and $t_{j}$, respectively, 𝜛 represents the system noise with Gaussian distribution, e represents the quantization bit number of the intelligent reflection surface, and $\theta_{n}$ represents the phase shift of the nth reflection module, and its desirable range is the discrete value with $\frac{\pi }{2^{e-1}}$ interval as an equal interval on $\left[0,2\pi \right]$. Thus, $\theta_{n}=\frac{y\pi }{2^{e-1}},\;0\leq y\leq 2^{e}$. The numerator of Equation (1) represents the useful signal power, and the denominator represents the interference from other transmitters $t_{j}$ and noise.

Correspondingly, the data rate $R_{i}$ of user i can be expressed as
(2)$$R_{i}=\log_{2}\left(1+\frac{{\left|h_{r_{i},t_{i}}+\sum_{n=1}^{N}g_{r_{i},t_{i}}^{n}e^{j\theta_{n}}\right|}^{2}p_{i}}{\sum_{j=1,j\neq i}^{I}{\left|h_{r_{i},t_{j}}+\sum_{n=1}^{N}g_{r_{i},t_{j}}^{n}e^{j\theta_{n}}\right|}^{2}p_{j}+𝜛}\right)$$

According to Equation (2), it can be concluded that the sum rate of users is
(3)$$R=\sum_{i=1}^{I}R_{i}$$

Therefore, the optimization problem can be expressed as
(4)$$max\;R=\underset{p_{1},\ldots ,p_{I,\theta_{1},\ldots ,\theta_{N}}}{\max}\sum_{i=1}^{I}\log_{2}\left(1+\frac{{\left|h_{r_{i},t_{i}}+\sum_{n=1}^{N}g_{r_{i},t_{i}}^{n}e^{j\theta_{n}}\right|}^{2}p_{i}}{\sum_{j=1,j\neq i}^{I}{\left|h_{r_{i},t_{j}}+\sum_{n=1}^{N}g_{r_{i},t_{j}}^{n}e^{j\theta_{n}}\right|}^{2}p_{j}+𝜛}\right)\;subject\;to\;C1:0<p_{i}\leq p_{max},0<i<I\;C2:R_{i}\geq R_{min},0<i<I\;C3:\theta_{n}=\frac{y\pi }{2^{e-1}},0\leq y\leq 2^{e},0<n<N$$
where constraint C1 means that the transmitted power cannot exceed the maximum limited power $p_{max}$; constraint C2 means that the rate of users should be greater than $R_{min}$; constraint C3 means that the amplitude of each element is 1, and the phase shift is a discrete quantity on the interval $\left[0,\;2\pi \right]$. As observed, the considered problem (4) is nonlinear and non-convex, involving the joint optimization of phase shift and power allocation. Hence, it is hard to solve directly. To deal with this problem, the joint optimization problem is usually decomposed into multiple sub-problems to be solved separately. However, these techniques are computationally complex, which leads to unnecessary delays in updating the optimal solution.

## 3. Joint Optimization Algorithm of Power and Phase Shift Based on PSO

In order to solve the proposed resource allocation problem more efficiently with lower complexity without degrading the system performance, we investigate low-complexity solutions in this section. Inspired by the advantages of good convergence performance and low complexity of the PSO algorithm, we propose the PSO-based power and phase shift allocation scheme.

### 3.1. Basic Concept of PSO

PSO is a classical swarm intelligence algorithm, which was proposed in [18]. The PSO algorithm uses particles with attributes instead of individuals in the flock to simulate the predation scene of the flock: in an open field, food is distributed in different places. Birds in different areas do not know where the food is, and they search to find the most food for the flock. The key to solving the foraging problem of particle swarm optimization lies in the information sharing and learning of individuals and groups. Each particle has two attributes of position and velocity. Flocks of birds fly within feeding range, changing position and velocity at any time. The adjustment of attributes at any given moment takes into account not only its own historical foraging, but also the properties of the particles closest to the most food in the current population. After a certain period of adjustment and flight, the particle swarm will gradually gather to the position of the most food.

The above predator-prey scenario can be abstracted into a mathematical problem, and the PSO algorithm solves this kind of maximum/minimum problem with multiple variables. The velocity of the particles represents the velocity of movement, and the position represents the direction of movement. The particles look for the optimal solution in the feasible space and mark it as the individual extreme value; all the particles in the population will disclose their search results, and the best of all individual extreme values will be marked as the global extreme value; in a certain period of time, the global extreme value almost no longer changes, which means that the PSO algorithm has finished running and has obtained the global optimal solution of the corresponding problem.

### 3.2. The Process of PSO Algorithm

The PSO algorithm will randomly generate the position and velocity of particles at the beginning, and then iterate to get the optimal solution [19,20,21,22,23,24]. In each iteration, the particle updates its velocity and position by tracking two extremes until the upper limit of the number of iterations is reached. The main flow of the algorithm (Algorithm 1) is as follows:
**Algorithm 1.** PSO algorithm.Step 1: InitializationInitialize the population size (the number of particles in the population), the number of iterations, the effective position range, the effective velocity range, and the initial position and velocity of each particle.
Step 2: Evaluate the fitness of each particle according to the fitness functionThe fitness function is the objective function of the algorithm optimization, and the function value calculated by bringing the attributes of the particle into the fitness function is the fitness of the particle.Step 3: Find pbest and gbestFor each particle, its current fitness is compared with the fitness corresponding to its individual historical best position (pbest). If the current fitness value is higher, the individual historical best position is updated to the current position.For each particle, its current fitness is compared with the fitness corresponding to the global best position (gbest). If the current fitness is higher, the global best position is updated to the current particle position.Step 4: Update particle attributeUpdate the velocity and position of each particle according to the update formula [19].The Common update formulas are as follows:$v_{i}\left(t+1\right)=v_{i}\left(t\right)+c_{1}\times rand()\times \left(pbest_{i}\left(t\right)-x_{i}\left(t\right)\right)+c_{2}\times rand()\times \left(gbest\left(t\right)-x_{i}\left(t\right)\right)$$x_{i}\left(t+1\right)=x_{i}\left(t\right)+v_{i}\left(t+1\right)$where $v_{i}\left(t\right)$ represents the velocity of particle i at t time, $x_{i}\left(t\right)$ represents the position of particle i at t time, $v_{i}\left(t+1\right)$ is the velocity of the particle at the next time, and $x_{i}\left(t+1\right)$ is the position of the particle at the next time. Particle $pbest_{i}\left(t\right)$ is the individual historical optimal value of particle i at t time, $gbest\left(t\right)$ is the global optimal value at t time, $c_{1}$ and $c_{2}$ are learning factors, and particle rand() denotes a random number between 0 and 1.If the iteration is complete, the algorithm stops; otherwise return to step 2.Notice that searching the maximum total rate for the IRS-assisted D2D communication system is similar to the process of obtaining the global optimal location in the particle swarm optimization algorithm. Therefore, this section will design the PSO algorithm for the IRS-assisted D2D underlaying cellular communication systems to optimize the phase shift and power.

### 3.3. The Proposed Power and Phase Shift Joint Optimization Algorithm

In order to solve the above optimization problem, the power and phase shift are compared to the particle position in the particle swarm optimization algorithm, and the fitness function, update formula, and update scheme are designed in detail. Through iterative optimization, the global optimal solution of the particle is found.

#### 3.3.1. Fitness Function

The optimization goal is to maximize the sum rate of the system, so the total rate calculation function is set to the fitness function; however, because of the constraints of the optimization objective, the fitness function needs to be adjusted. When dealing with the constraint problem, the penalty term is usually added to the objective function to realize the transformation from the constraint problem to the unconstrained problem. For the IRS-assisted D2D communication system, the total rate is constrained by the lowest rate of the user, so the corresponding penalty term should be designed for the fitness function.

The penalty term satisfies the constraint by eliminating the individual solution. In the problem of maximizing the objective function, when the value of the individual adaptation is large, but not within the constraint, the penalty term is added to reduce the fitness value, and thus the individual is eliminated. Specifically, if there are I constraints in the function $F\left(X\right)$ ($R_{i}\left(X\right)\geq R_{min},0<i<I$), inequality constraints are usually transformed into $R_{i}\left(X\right)-R_{min}\geq$0. At the same time, the I constraints are normalized, and the corresponding penalty term is transformed into $\sigma \sum_{i=1}^{I}max\{0,R_{min}-R_{i}\left(X\right)\}$. Here, σ is the punishment factor.

For the total rate optimization problem, the constraint $R_{i}\geq R_{min}$ needs to be satisfied, and the penalty term is designed as
(5)$$G\left(X\right)=\frac{R\left(X\right)}{R_{min}}\frac{\sum_{i=1}^{I}max\{0,R_{min}-R_{i}\}}{I}$$
where X is the current location of any individual, and $R\left(X\right)$ is the sum of the user rates corresponding to the current location. Then the fitness function is
(6)$$Fitness\left(X\right)=R\left(X\right)-G\left(X\right)$$

As observed, $max\{0,R_{min}-R_{i}\}=0$ if $R_{i}\geq R_{min}$. If all constraints are satisfied, $Fitness\left(X\right)$ is the objective function without penalty.

#### 3.3.2. Update Formula

The particle swarm has M particles, and the dimension of each particle $X_{m}=\left[p_{1},\ldots ,p_{I,}\theta_{1},\ldots ,\theta_{N}\right]$ is I+N, that is, the first I dimension of the particle position represents the user power, and the latter N dimension represents the component phase shift. In order to improve the search ability, the particle velocity should have inertia. Generally speaking, a larger inertia weight is beneficial to the global search, while a smaller weight is more conducive to the local search. Here, the weight factor is set to $w=w_{max}-\left(w_{max}-w_{min}\right)*{\left(t/T\right)}^{2}$, where $w_{min}$ and $w_{max}$ are the minimum and maximum inertia weight, respectively [21]; T represents the total number of iterations, and w decreases with the increase of the number of iterations, so that the particle search has a strong global search ability in the early stage and a stronger local search ability in the later stage of iteration. For the *m*th particle, the update formulas of its velocity $V_{m}$ and position $X_{m}$ at the t iteration are as follows:(7)$$V_{m}\left(t+1\right)=w\times V_{m}\left(t\right)+c_{1}\times r\times \left(pbest_{m}\left(t\right)-X_{m}\left(t\right)\right)+c_{2}\times r\times \left(gbest\left(t\right)-X_{m}\left(t\right)\right)$$
(8)$$X_{m}\left(t+1\right)=X_{m}\left(t\right)+V_{m}\left(t+1\right)$$
where $pbest_{m}\left(t\right)$ denotes the individual historical optimal solution of any particle *m* in the t iteration, $gbest\left(t\right)$ denotes the global optimal solution in the t iteration, w is the inertia weight factor, $c_{1}$ and $c_{2}$ represent the learning factor, and r represents the random number between 0 and 1. $V_{m}$ and $X_{m}$ are divided into two parts, the first I columns are marked as matrix $V_{m}^{I}$ and $X_{m}^{I},$ respectively, indicating the corresponding velocity and position of the power scheme, and the last N columns are marked as matrix $V_{m}^{N}$ and $X_{m}^{N}$, respectively, indicating the corresponding velocity and position of the phase shift scheme.

#### 3.3.3. Penalty-First Update Scheme

The velocity and position of each particle should meet the constraint of (4), so after calculating the new position and velocity according to the updated formula, the particle attributes need to be corrected first.

■Update the velocity of particles

The first I dimension of particle velocity should meet: $v_{pmin}\leq V^{I}\leq v_{pmax}$, and the latter N dimension should meet: $v_{tmin}\leq V^{N}\leq v_{tmax}$, where $v_{pmax}$ and $v_{pmin}$ represent the upper limit and lower limit of power velocity, respectively. $v_{tmax}$ and $v_{tmin}$ represent the upper and lower limit of phase shift velocity, respectively.

Then, update directly when the particle velocity meets the constraint; otherwise, set it to the corresponding boundary value, for example, if the velocity of the first *I* dimension is greater than $v_{pmax}$, update it to $v_{pmax}$.

■Update the position of particles

The first I dimension of the particle position corresponds to the user power, so it only needs to satisfy the constraint: $x_{pmin}<x\leq x_{pmax}$, where $x_{pmin}$ and $x_{pmax}$ represent the upper and lower limits of the user power, respectively. The last N dimension of the particle position corresponds to the component phase shift, which should satisfy the constraint $x_{tmin}\leq x\leq x_{tmax}$, where $x_{tmin\;}\mathrm{and}\;x_{tmax}$ represent the upper and lower limits of the component phase shift, respectively. Because the phase shift is a discrete quantity with an equal interval, and the position value calculated by the update formula is continuous, the corresponding update scheme needs to be improved.

Therefore, we adjust the velocity and position of each particle to meet the discrete value constraint of the phase shift. First, for any particle m, use the sigmiod function to adjust all velocity values in $V_{m}^{N}$ to a number between 0 and 1, and generate a random number between 0 and 1. For the velocity $v_{n}$ corresponding to any column *n* in $V_{m}^{N}$, if the adjustment is greater than r, then the position $x_{n}$ corresponding to the nth column in the $X_{m}^{N}$ is adjusted to be greater than or equal to the nearest preferred phase shift dispersion value of $x_{n}$; otherwise, it is less than or equal to the nearest preferred phase shift dispersion value of $x_{n}$.

■Updates of pbest and gbest

For the individual historical optimal solution of any particle m, $pbest_{m}=X_{m}\left(t\right)$, if any of the following conditions are satisfied, it will be updated to the current solution of the particle:

The penalty value of the current solution of the particle is less than the penalty value of the historical optimal solution of the particle. That is,


(9)
$$G\left(X_{m}\left(t\right)\right)<\underset{i=1,\ldots .,t-1}{\min}G\left(X_{m}\left(i\right)\right)$$


2.The penalty value of the current solution of the particle is equal to the penalty value of the individual historical optimal solution of the particle, and the fitness value is greater than that of the individual historical optimal solution. That is,


(10)
$$G\left(X_{m}\left(t\right)\right)=\underset{i=1,\ldots ,t-1}{\min}G\left(X_{m}\left(i\right)\right),\;and\;Fitness\left(X_{m}\left(t\right)\right)>\underset{i=1,\ldots ,t-1}{\max}Fitness\left(X_{m}\left(i\right)\right)$$


For the population optimal solution denoted by $gbest=X_{m0}$, if any of the following conditions are satisfied, it will be updated to the individual solution of the current particle:

The current penalty value of the individual particle solution is less than that of the population optimal solution. That is, for the current time *t*, and individual particle m0


(11)
$$G\left(X_{m0}\right)<\underset{\begin{array}{c} m=1,\ldots .,M\\ m\neq m0 \end{array}}{\min}G\left(X_{m}\right)$$


2.The penalty value of the current particle individual solution is equal to the penalty value of the population optimal solution, and the fitness value is greater than the fitness value of the population optimal solution. That is,


(12)
$$G\left(X_{m0}\right)=\underset{\begin{array}{c} m=1,\ldots .,M\\ m\neq m0 \end{array}}{\min}G\left(X_{m}\right),\;and\;Fitness\left(X_{m0}\right)>\underset{\begin{array}{c} m=1,\ldots .,M\\ m\neq m0 \end{array}}{\max}Fitness\left(X_{m}\right)$$


Based on the above analysis, the PSO-based joint optimization steps are described in Figure 2.

According to the steps, we summarize the PSO-based joint optimization algorithm (Algorithm 2) for phase shift and power as follows:
**Algorithm 2.** Power and Phase Shift Joint Optimization Algorithm Based on PSO.Initialization: randomly generate the initial velocity and position of the population, set pbest and gbest
for *t* = 1 to *T* do;for *m* = 1 to *M* do; Calculate the weight factor w; Calculate the fitness of particles according to (6); Update the velocity and position of particle *m* according to the updated Formulas (7) and (8); Update the individual history optimal solution and population optimal solution according to the penalty-first update scheme in Section 3.3.3;End for;If the number of iterations is less than *T*, then t=t+1, return 3; if the number of iterations reaches *T*, the iteration ends and the optimal solution of the population is output;End for.

### 3.4. Complexity Analysis

The complexity of the joint optimization algorithm of the phase shift and power allocation based on the PSO algorithm mainly depends on the number of iterations and the number of particles. The algorithm terminates when the number of iterations reaches *N*, and *M* particles need to be updated in each iteration. The computational complexity of the update process is constant order, so the complexity of the algorithm is *O*(*MT*). The complexity of the algorithm proposed in this paper is much lower than that of the alternating iterative optimization algorithm proposed in [17].

## 4. Simulation Results and Analysis

To evaluate the performance of the proposed PSO-based optimization algorithm, we provide simulations and analysis in this section. The specific settings of the simulation scenario (shown in Figure 3) are as follows: The base station is located at the origin, IRS is placed on the *Y-Z* plane, the spacing between each element square is 0.05 m, and the *X*-axis is the axis of symmetry. One CUE and I−1 pairs of DUE are randomly distributed in the simulation region where the absolute values of the horizontal and vertical coordinates are both greater than 20. The specific simulation parameters are shown in Table 1.

In order to verify the effectiveness of the algorithm proposed in this paper, we compared it with the PSO algorithm described in [21], the alternating iterative optimization algorithm proposed in [17], and the distributed optimization PSO algorithm. The main idea of the distributed optimization PSO algorithm is as follows: we divide the phase shift selection and power control into two sub-problems. Firstly, the PSO algorithm is used to optimize the phase shift under the condition of fixed phase shift, and then the power is fixed to complete the PSO-based phase shift optimization.

Figure 4 depicts the comparison of the sum rate of different algorithms as a function of the number of D2D users. It can be seen from the figure that the total system rate is improved as the number of D2D users increases. Among them, the local search algorithm of alternating iteration proposed in [17] has the best optimization effect, and the algorithm proposed in this paper is similar to it. When the number of D2D users is four, the total rate of the system achieved by our algorithm is 1.8% lower than that of the alternating iterative algorithm in [17]. As observed, compared with the PSO and distributed PSO, the sum rate of the proposed algorithm increases by about 10.2% and 38.3%, respectively, when I = 5. The performance of the algorithm in [21] is limited because the phase shift of the components in the system is not optimized specifically. With the increase of the number of users, the negative impact caused by the reflected interference in the network becomes more and more serious, and the sum rate of the system gradually flattens out.

In this paper, the proposed algorithm achieves lower power consumption on the basis of the performance approaching the alternating iterative optimization algorithm. As shown in Figure 5, the algorithm in [21] only carries out targeted optimization of user power, so although the performance is limited, the power consumption is the lowest. In terms of energy efficiency, the proposed algorithm is obviously superior to the alternate iterative optimization algorithm, and the total power consumption has an advantage of 20% when the number of D2D users is four.

The iteration times of PSO algorithm directly affect the optimization effect. Figure 6 shows that as the number of iterations increases, the performance increases first and then becomes stable. Specifically, the simulation is carried out when the number of D2D users is two, four, and six, respectively. When iteration times of the algorithm are 20, the total rate of the system achieved is relatively low, while at higher iteration times, the total rate of the system that can be reached increases to a certain extent. In particular, when there are only three users in the system, the algorithm with 180 iterations outperforms 20 iterations by 11.8% in performance. Therefore, different iteration times should be designed according to different optimization requirements. Generally, increasing the number of iterations improves the sum rate, and reducing the number of iterations reduces the running time of the algorithm.

Figure 7 shows the results of the fitness value of the particles with the optimal strategy in the population as a function of the number of iterations. As shown in the figure, the fitness of the optimal individual increases rapidly when the number of iterations is less than 50, which verifies that the proposed joint optimization algorithm has good convergence performance. When the number of iterations exceeds 50, the fitness value increases in steps. Among them, the long-term flatness is because the search is trapped in the local optimal, and the population cannot find a better strategy temporarily. However, when the fitness value changes, it will show a trend of leaping in a small range. This is because of the randomness brought by the update scheme using the sigmoid function for discrete value constraints when the phase shift optimization variable is updated. Therefore, in the late stage of population evolution, the system still maintains a certain ability to break through the local optimal, making the total rate of the system as close as possible to the optimal scheme.

## 5. Conclusions

In order to reduce the complexity of the phase shift selection and power allocation algorithm, this paper proposes a joint optimization algorithm of phase shift and power allocation based on multi-objective PSO. Specifically, the constraint conditions of system problems are transformed into penalty terms, and a unique updating scheme is designed for the discrete value constraint of the phase shift. In addition, all particles in the population evolve under the constraint of position and velocity, and finally obtain the optimal particle strategy. Simulation results show that, compared with the other three algorithms, the proposed algorithm has the best performance in improving the system sum rate and reducing power consumption.

## 6. Future Research

The algorithm proposed in this paper effectively improves the performance of the IRS-D2D communication system, but there are still some imperfections, which can be further studied from the following aspects in the future.

(a)Study the number and location of IRS and analyze the impact of these factors on performance to further explore the potential of IRS-assisted D2D communication systems;(b)There are many different domains where advanced optimization algorithms have been applied as solution approaches, such as online learning, multi-objective optimization, and others. In our future work, we will explore advanced optimization algorithms (e.g., adaptive heuristics and metaheuristics, self-adaptive algorithms, diffused algorithms, island algorithms, etc.) [25,26,27]. Moreover, the PSO approach adopted in this study can be compared with these advanced optimization algorithms;(c)In this paper, we only consider the radio resource allocation when the channel state information (CSI) is perfect. In future work, the optimization problems under the condition of imperfect CSI should also be considered.

## Figures and Tables

**Figure 1 sensors-23-05266-f001:**
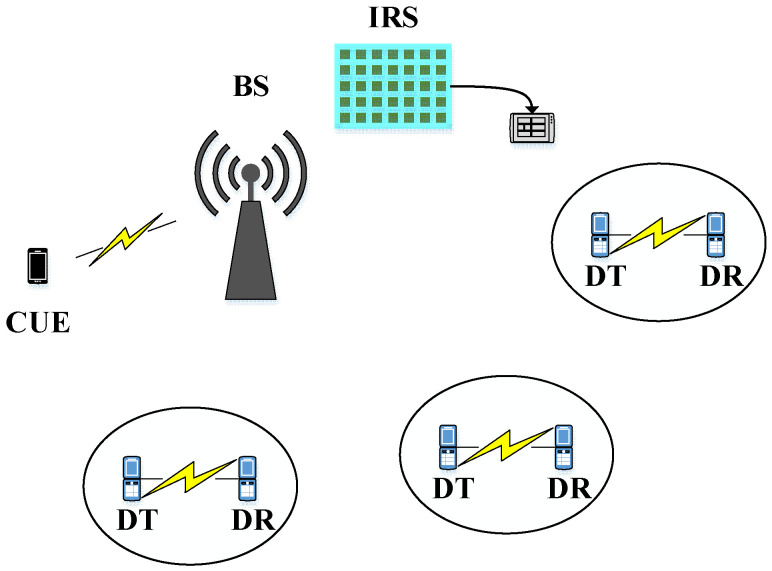
D2D communication system with single cellular users and multiple co-channel D2D users.

**Figure 2 sensors-23-05266-f002:**
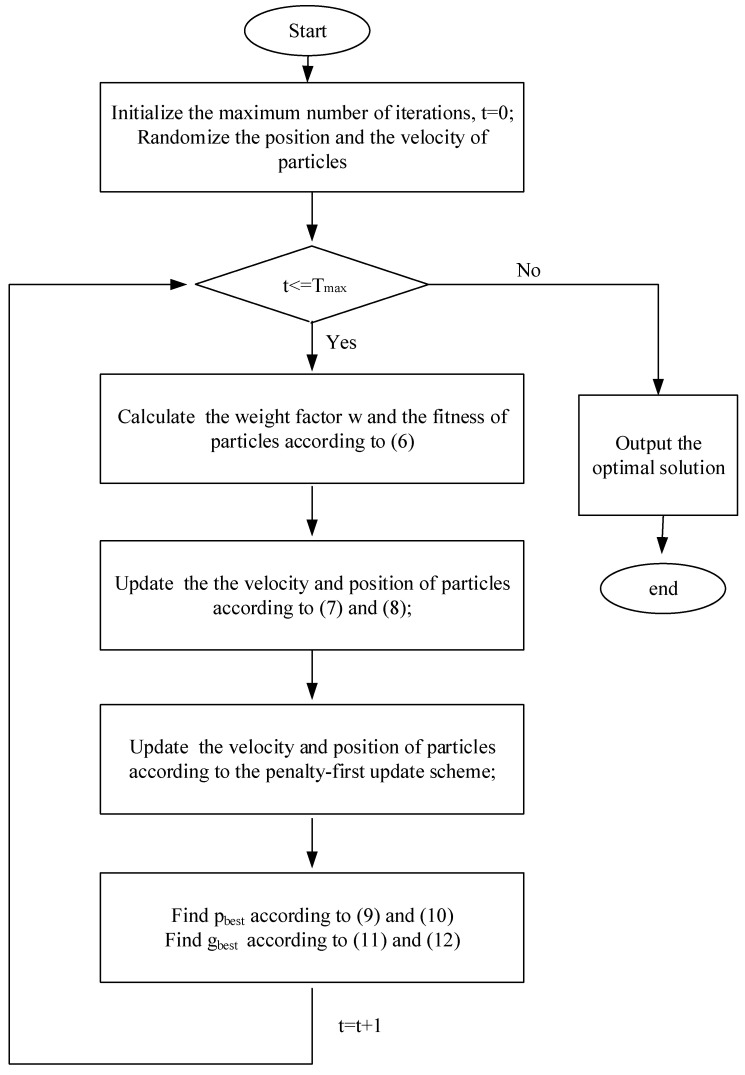
Flowchart of the power and phase shift allocation scheme based on PSO.

**Figure 3 sensors-23-05266-f003:**
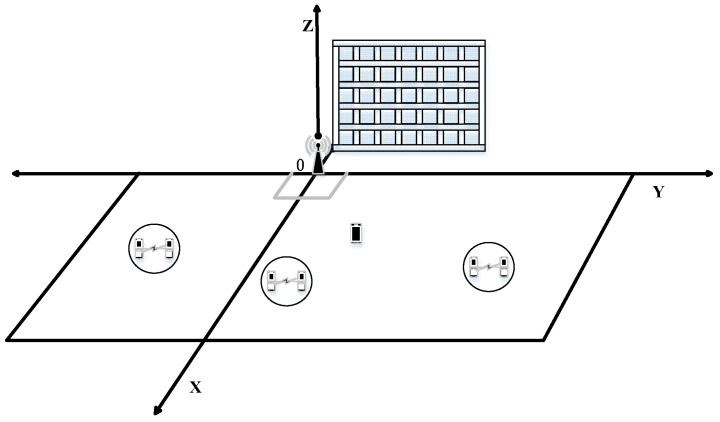
Simulation scenario.

**Figure 4 sensors-23-05266-f004:**
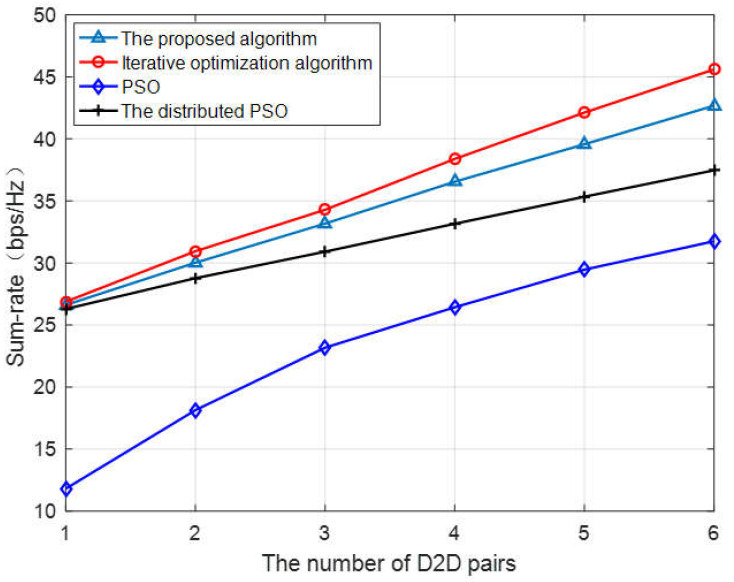
Sum rate vs. the number of D2D users.

**Figure 5 sensors-23-05266-f005:**
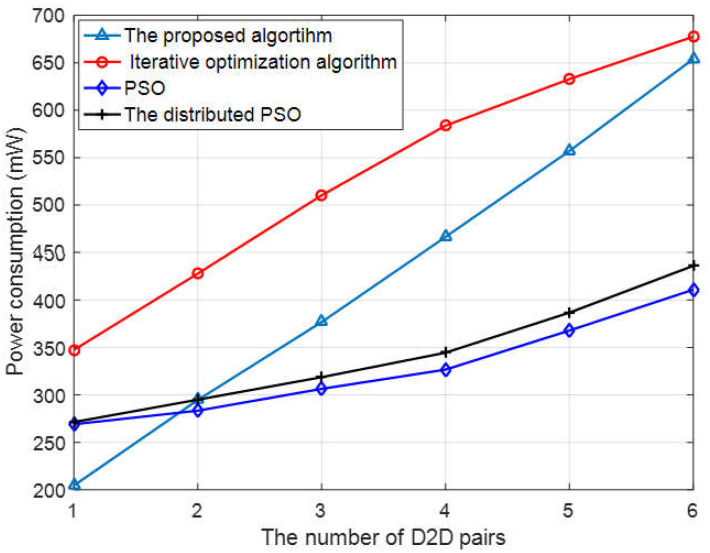
The power consumption vs. the number of D2D users.

**Figure 6 sensors-23-05266-f006:**
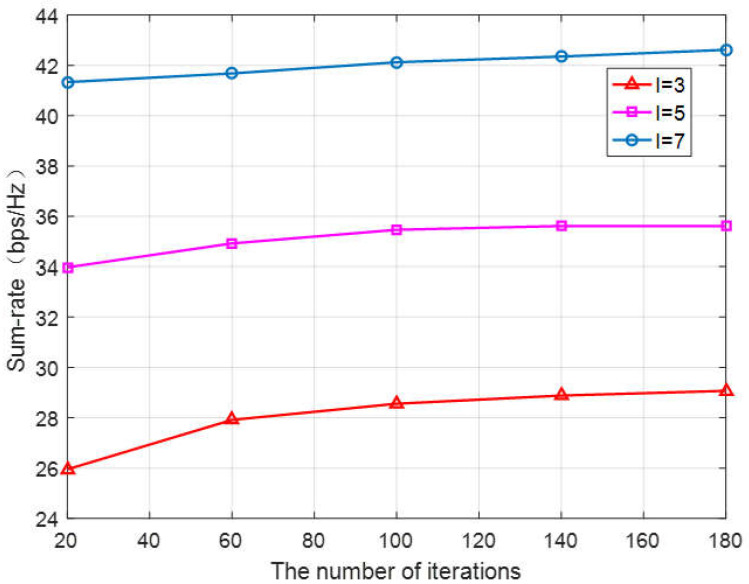
The sum rate vs. the number of iterations with different D2D users.

**Figure 7 sensors-23-05266-f007:**
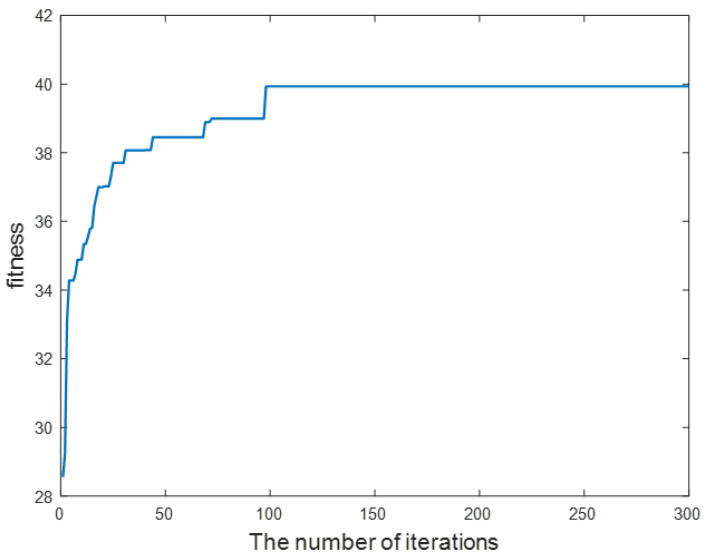
The optimal individual fitness vs. the number of iterations.

**Table 1 sensors-23-05266-t001:** Parameter values used in the simulation.

Symbol and Meaning	Value
Number of users I	2:1:7
Number of IRS elements N	16
Number of bits quantized by the IRS element e	4
Bandwidth of the system	28 GHz
Noise power spectral density	−134 dBm/Hz
Number of Population particle *M*	80
Iteration times T	200
Upper power limit of the users $p_{max}$	200 mW
Inertia weight factor $w_{max}$	0.9
Inertia weight factor $w_{min}$	0.4
The minimum rate of the users $R_{min}$	1.58 bps/Hz
Learning factor $c_{1},c_{2}$	1.49445
$v_{pmin}$	−10
$v_{pmax}$	10
$v_{tmin}$	−π/8
$v_{tmax}$	π/8

## Data Availability

Not applicable.

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
