# Peer review of "Joint Particle Swarm Optimization of Power and Phase Shift for IRS-Aided D2D Underlaying Cellular Systems"

_sensors, 2023, doi:10.3390/s23115266_

Round 1
Reviewer 1 Report
Paper sensors-2365552 “Joint particle swarm optimization of power and phase shift for
IRS-aided D2D underlaying cellular systems”
Comments
This study investigates the Joint particle swarm optimization of power and phase shift for IRS-aided D2D underlaying cellular systems. I think the paper fits well the scope of the journal and addresses an important subject. However, a number of revisions are required before the paper can be considered for publication. There are some weak points that have to be strengthened. Below please find more specific comments:
*Abstract: The abstract seems kind of short. Please try to make it more detailed and highlight better the findings from this research.
*Keywords: The keywords seem to be adequate. No comments.
*The literature review seems kind of short. Please double check for the most recent and relevant studies published over the last 2-3 years.
*Please make sure that all the adopted assumptions are supported by the relevant references. This will help justifying the adoption of these assumptions.
*The presentation of the proposed methodology could be more detailed. I suggest including the flowchart with the main algorithmic steps and discuss every stem of the algorithm in more detail.
*Please provide more details regarding the input data used throughout this study. More supporting references would be helpful to justify the data selection.
*The manuscript contains quite a lot of figures and tables. Please double check and try to provide a more detailed description of these figures and tables where appropriate to make sure that the future readers will have a reasonable understanding of what these figures and tables represent.
*Future research: The authors primarily used PSO for the studied decision problem. In the future research, the authors should explore advanced optimization algorithms for this decision problem. The authors should create a general discussion regarding the importance of advanced optimization algorithms (e.g., adaptive heuristics and metaheuristics, self-adaptive algorithms, diffused algorithms, island algorithms, etc.) for challenging decision problems. There are many different domains where advanced optimization algorithms have been applied as solution approaches, such as online learning, scheduling, multi-objective optimization, transportation, medicine, data classification, and others (not just the decision problem addressed in this study). The authors should create a discussion that highlights the effectiveness of advanced optimization algorithms in the aforementioned domains and their potential applications for the decision problem addressed in this study. The PSO approach adopted in this study can be compared with these advanced optimization algorithms as a part of the future research. This discussion should be supported by the relevant references, including but not limited to the following:
Hybrid gene selection approach using XGBoost and multi-objective genetic algorithm for cancer classification. Medical & Biological Engineering & Computing 2022, 60(3), pp.663-681.
Berth scheduling at marine container terminals: a universal island-based metaheuristic approach. Maritime Business Review 2020, 5 (1), pp.30-66.
Preventive maintenance for the flexible flowshop scheduling under uncertainty: A waste-to-energy system. Environmental Science and Pollution Research 2021, pp.1-20.
Such a discussion will help improving the last section of the manuscript significantly.
Reviewer 2 Report
Please find the comments in the attached file.

The quality of English must be improved. Please find some details in the attached file.
Reviewer 3 Report
Dear Authors,
I will be interesting to observer the following points:
a) Abstract and Introduction:
- The acronym D2D is required to be explained in the first line inside the abstract and introduction sections. D2D (Device-to-device) (D2D) communication.
- A direct and clear statement saying that "In this research/work/paper the contribution is...." It will improve the message of the contribution.
- In the end of the abstract there is a statement, that it will be interesting to present some numbers: "The simulation results show...."
- Inside the Introduction section, it is essential a paragraph directly presenting which is the focus of the problem tackle, environment, and results. It is also desirable the structure of the paper, and why the section selection.
b) Section 2: again, do not start without explaining why the choices are presented.
c) Section 3: very (very) important to begin this section with original reasons to adopt these approaches. Write a initial paragraph, before section 3.1 drawing a link between the present research and those topics of section 3.
d) Section 4: The first sentence of this section is:
"The specific settings of the simulation scenario are as follows:" the obvious question from a reader is why? what? It vital a preface of one clear sentence explain why aspect of this important section from the research.
e) Section 4: It is strongly suggested a future work paragraph indicating some direction of future efforts.
It will be interesting a English review, because some parts from the contribution are coloquial.
Round 2
Reviewer 1 Report
The authors have adequately addressed my original concerns regarding the manuscript. The quality and presentation of the manuscript have been improved. Therefore, I recommend acceptance.
Reviewer 2 Report
Previous concerns have been well addressed.
Reviewer 3 Report
The paper now has another approach in which a reader can capture the entire message from your research.
It is always available to provide a more clear and objetive contribution, in which the paper is inside this region.
It is necessary a more carefully paper read to fix some smalls points. I will be interesting to an external member from your group to provide his/her view in terms of English understanding